# Evidence for the Desmosomal Cadherin Desmoglein-3 in Regulating YAP and Phospho-YAP in Keratinocyte Responses to Mechanical Forces

**DOI:** 10.3390/ijms20246221

**Published:** 2019-12-10

**Authors:** Jutamas Uttagomol, Usama Sharif Ahmad, Ambreen Rehman, Yunying Huang, Ana C. Laly, Angray Kang, Jan Soetaert, Randy Chance, Muy-Teck Teh, John T. Connelly, Hong Wan

**Affiliations:** 1Centre for Oral Immunobiology and Regenerative Medicine, Institute of Dentistry, Barts and The London School of Medicine and Dentistry, Queen Mary University of London, London E1 2AT, UK; j.uttagomol@qmul.ac.uk (J.U.); u.s.ahmad@qmul.ac.uk (U.S.A.); a.rehman@qmul.ac.uk (A.R.); yunying.huang@qmul.ac.uk (Y.H.); a.s.kang@qmul.ac.uk (A.K.); r.chance@qmul.ac.uk (R.C.); m.t.teh@qmul.ac.uk (M.-T.T.); 2Centre for Cell Biology and Cutaneous Research, Blizard Institute; Barts and The London School of Medicine and Dentistry, Queen Mary University of London, London E1 2AT, UK; a.c.lalyaguedo@qmul.ac.uk (A.C.L.); j.soetaert@qmul.ac.uk (J.S.); j.connelly@qmul.ac.uk (J.T.C.)

**Keywords:** desmoglein, desmosome, adherens junction, YAP, Phospho-YAP, keratinocyte, cyclic strain, substrate stiffness

## Abstract

Desmoglein 3 (Dsg3) plays a crucial role in cell-cell adhesion and tissue integrity. Increasing evidence suggests that Dsg3 acts as a regulator of cellular mechanotransduction, but little is known about its direct role in mechanical force transmission. The present study investigated the impact of cyclic strain and substrate stiffness on Dsg3 expression and its role in mechanotransduction in keratinocytes. A direct comparison was made with E-cadherin, a well-characterized mechanosensor. Exposure of oral and skin keratinocytes to equiaxial cyclic strain promoted changes in the expression and localization of junction assembly proteins. The knockdown of Dsg3 by siRNA blocked strain-induced junctional remodeling of E-cadherin and Myosin IIa. Importantly, the study demonstrated that Dsg3 regulates the expression and localization of yes-associated protein (YAP), a mechanosensory, and an effector of the Hippo pathway. Furthermore, we showed that Dsg3 formed a complex with phospho-YAP and sequestered it to the plasma membrane, while Dsg3 depletion had an impact on both YAP and phospho-YAP in their response to mechanical forces, increasing the sensitivity of keratinocytes to the strain or substrate rigidity-induced nuclear relocation of YAP and phospho-YAP. Plakophilin 1 (PKP1) seemed to be crucial in recruiting the complex containing Dsg3/phospho-YAP to the cell surface since its silencing affected Dsg3 junctional assembly with concomitant loss of phospho-YAP at the cell periphery. Finally, we demonstrated that this Dsg3/YAP pathway has an influence on the expression of *YAP1* target genes and cell proliferation. Together, these findings provide evidence of a novel role for Dsg3 in keratinocyte mechanotransduction.

## 1. Introduction

The hallmark of epithelia is that individual cells attach to each other through numerous intercellular junctions with two anchoring junctions, i.e., adherens junctions (AJs) and desmosomes (DSMs), being especially important for cellular architecture and tissue integrity. The cell-cell adhesions in both junctions are mediated by cadherin family proteins in a calcium-dependent manner. The adhesion receptors in AJs belong to the classical cadherin family, such as E-cadherin, whereas those in DSMs are comprised of two desmosomal cadherin subfamilies, desmoglein (Dsg) and desmocollin (Dsc). All these cadherins are the single-pass transmembrane proteins and are linked to cytoskeletal filaments on their cytoplasmic tails through plaque proteins, including armadillo proteins (α, β, γ-catenins, and plakophilins) and plakin family members (e.g., desmoplakin). Both AJs and DSMs are enriched in stratified epithelial tissues, such as skin and mucous membranes that are subjected to mechanical stresses on a daily basis.

Cadherin-mediated cell adhesion plays an essential role in contact inhibition of cell proliferation (CIP) via the Hippo signaling pathway—the latter being an essential regulator in control of cell differentiation and organ growth, with its deregulation contributing to cancer development [1,2]. The depletion of the components in the Hippo pathway, such as β- and α-catenins, inhibits the E-cadherin mediated CIP, leading to increased cell proliferation [3]. This is due to the activation of the Yes-associated protein (YAP), a transcriptional coactivator, and a key downstream nuclear effector of the Hippo pathway with its activation causing YAP phosphorylation at specific serine residues, such as serine 127 (S127), resulting in nuclear exclusion [3]. YAP has also been identified as a mechanosensor independent of the Hippo cascade, and such a function is driven by Rho GTPase activity and the tension generated by the actomyosin cytoskeleton [4,5]. The external physical forces, such as mechanical loading and substrate stiffness, can activate YAP and drive its nuclear translocation, leading to transcription of YAP target genes responsible for cell proliferation [5,6]. It is increasingly appreciated that mechanotransduction is as important as chemical factors in controlling diverse cell behavior, including growth, differentiation, and tumor progression [7,8].

E-cadherin-mediated AJs, in concert with associated filamentous actin (F-actin) and actomyosin, serve as important hubs in mechanosensing and mechanotransduction that influence cell-cell adhesion and traction force at cell-extracellular matrix (ECM) junctions [7,9,10,11,12,13]. However, little is known about the role of desmosomal cadherins in these processes in spite of the fact that DSMs are crucial in cell signaling and tissue integrity [14]. The importance of desmosomal cadherins, as well as other DSM constitutive proteins, in tissue homeostasis and physiological functions, has been underscored by various vesiculobullous diseases with a clinical manifestation of blistering and fragility syndrome of skin and oral mucosa, as well as heart failure in some cases [14,15]. Recent studies have uncovered that two DSM proteins, desmoglein-2 (Dsg2) and desmoplakin (Dp), are capable of sensing external mechanical loading, highlighting the probable involvement of DSM proteins in mechanobiology [16,17].

Desmoglein-3 (Dsg3) is an isoform of the Dsg subfamily with uniform expression across the entire stratified epithelium in oral mucosa and restricted expression in the basal layer of the epidermis in the skin [18]. Why Dsg3 exhibits such distinct tissue expression patterns remains unclear. Perhaps, Dsg3 is best known as PVA, the autoantigen in pemphigus vulgaris, a life-threatening autoimmune blistering disease. The importance of Dsg3 in cell-cell adhesion is highlighted by many studies based on pemphigus autoantibodies that demonstrate convincingly that disruption of Dsg3 is a causative factor for the skin and oral lesions [19,20]. However, the action of Dsg3 is not restricted to DSM adhesion, and in fact, Dsg3 is present at the plasma membrane beyond the DSMs [19,20,21,22,23,24]. In vitro studies have shown that extra-junctional Dsg3 “cross-talks” with E-cadherin and regulates various signaling pathways, e.g., Src, Ezrin, and Rho GTPases Rac1/cdc42, and the transcription factor c-Jun/AP-1, all of which are involved in modulating the actin cytoskeleton, implying a role for Dsg3 in mechanotransduction [23,24,25,26]. Moreover, the interaction of actin proteins with the cytoplasmic domain of Dsg3 has been demonstrated by Mass Spectrometry analysis [25]. The overexpression of Dsg3 results in enhanced membrane projections and cell spreading, with a consequence of accelerated cell migration and invasion in cancer cell lines [23,25]. The upregulation of Dsg3 has also been found in solid cancer where tissue stiffness has been altered [27,28,29,30,31]. Collectively, these lines of evidence led us to hypothesize that Dsg3 may be involved in cellular force transmission and transduction, somewhat analogous to E-cadherin. In this present study, we have addressed this hypothesis in keratinocytes and found that Dsg3 responds to mechanical loading and is required for AJ remodeling induced by cyclic strain or substrate stiffness. Importantly, we have provided evidence that Dsg3 seems to act as a regulator of YAP and phospho-YAP (pYAP) in cell proliferation and in response to mechanical stress via at least one mechanism of forming a complex with pYAP and regulating its cellular localization.

## 2. Results

### 2.1. Mechanical Cyclic Strain Alters Junction Protein Expression and Distribution in Keratinocytes

To address our hypothesis that Dsg3 may respond to external mechanical loading, we performed the equiaxial strain assay with human skin-derived, spontaneously transformed aneuploidy immortal HaCaT keratinocyte line, and oral-mucosa derived SqCC/Y1 keratinocyte lines [18]. First, immunofluorescence was performed to examine the expression and distribution of junctional proteins in situ, including Dsg3, in cells with or without Triton X-100 permeabilization. Since E-cadherin and nonmuscle actomyosin have been identified as force sensors [7,9,32,33,34], we monitored the expression of E-cadherin and Myosin IIa alongside Dsg3 in both cell lines (Figure 1A). It was found that the peripheral protein analysis (-Triton) was more revealing and showed a marked increase of both cadherins and Myosin IIa in their response to cyclic mechanical strains, in particular, in SqCC/Y1 cells, compared to their static counterparts (Figure 1A). The peripheral protein expression of Dp and α-catenin also showed an evident increase in strained SqCC/Y1 cells (Appendix A). Compared to SqCC/Y1, fewer alterations were observed in HaCaT cells except for phospho-Myosin Light Chain (pMLC), which exhibited a clear rise in strained cells compared to stationary control (Appendix A). Next, Triton-soluble (cytoskeletal non-associated) and insoluble (cytoskeletal-associated) fractions were extracted separately and analyzed by Western blotting as described previously [23,24]. We detected strain-induced elevation of most junctional proteins, including E-cadherin and Dsg3, in both fractions in SqCC/Y1 cells and again, to a lesser extent, in HaCaTs (Figure 1B). Increased actin was also observed in the Triton-insoluble fraction of strained SqCC/Y1 cells that may be related to enhanced translocation of both cadherins to the junctions. Then, HaCaT cells were subjected to surface protein fractionation analysis by Western blotting that revealed elevated expression of Dp, E-cadherin, and Dsg3 in strained cells compared to non-strained counterparts (Figure 1C). Together, these results demonstrated that mechanical loading induces increased expression and surface assembly of junctional proteins, including Dsg3 and E-cadherin, with changes of protein expression more evident in oral keratinocytes than skin-derived HaCaT cells.

### 2.2. Dsg3 Is Required for E-cadherin and Actomyosin Junction Assembly in Response to Mechanical Strain

To address the importance of Dsg3 in response to cyclic strain, we performed a Dsg3-knockdown study with transient transfection of small interfering RNA (siRNA) (100 nM) in HaCaTs in conjunction with cyclic strain. Both quantitative reverse transcription-polymerase chain reaction (RT-qPCR) and Western blotting analyses were performed in cells harvested instantly after strain or no strain, validating significant knockdown of Dsg3 at the transcript and protein levels (Appendix A). No reduction for other desmosomal cadherins, including *DSG2*, was found at the transcript levels as well as their response to the strain, except for *DSP* that showed an increase in strained cells with Dsg3 depletion compared to its static counterpart (Appendix A). On the other hand, *DSC2*, as well as desmocollin-2 (Dsc2) protein, showed a significant increase following Dsg3 depletion compared to controls, suggesting its likely compensation for the loss of desmosomal cadherins, such as desmocollin-3 (Dsc3) and Dsg2 in addition to Dsg3 (Appendix A). Notably, plakophilin-1 (PKP1) displayed an evident decrease in the Triton insoluble fraction in knockdown cells. In addition, an increase of actin in the insoluble fraction with a diminution of its soluble counterpart was also observed in cells in response to strain (Appendix A). It is also worth noting that a minor increase of residual Dsg3 frequently was observed in the knockdown cells subjected to strain suggesting its sensitivity to mechanical loading as described above. Nevertheless, some variations were observed in different protein fractionation analyses in this study. Significantly, immunostaining for peripheral E-cadherin and Myosin IIa revealed their marked reduction, with disruption of E-cadherin at the junctions, in Dsg3 knockdown cells with exposure to cyclic strain (arrow Figure 2A,B), indicating that Dsg3 depletion had a negative impact on E-cadherin junction assembly in its response to mechanical loading. In support, Dsg3 overexpression (T8-D3) (gain-of-function) in cutaneous T8 cell line demonstrated enhanced E-cadherin and Myosin IIa at the junctions compared to control or T8-D3 cells with Dsg3 RNA interference (RNAi) that evoked disruption of junctions (Appendix A). Together, these results demonstrated that Dsg3 is required for junction assembly in keratinocyte response to mechanical loading.

DSMs confer strong cell-cell adhesion by forming calcium-independent junctions in confluent keratinocyte cultures [35]. Next, we analyzed the nature of DSMs in cells with Dp immunostaining [35]. The result showed numerous calcium-independent DSMs in HaCaT cells of all conditions. In control siRNA treated cells, the mechanical strain resulted in a more even and concentrated linear distribution of Dp at the junctions (arrowheads Figure 2C). In contrast, Dsg3 knockdown resulted in a punctate Dp staining pattern in non-strained cells but with a marked reduction in strained cells compared to the respective controls (arrows Figure 2C,D). This finding indicates a defect in the DSM assembly induced by cyclic strain in Dsg3 depleted cells and reinforces that Dsg3 is required for both AJ and DSM junction assembly in response to mechanical force. Furthermore, the protein turnover for various junctional proteins, including Dp, in response to strain for up to 24 h was analyzed and the result showed a gradual decline of Dp in response to strain compared to non-strained cells (Appendix A), indicating dynamic remodeling of the DSMs in response to mechanical loading. Subtle changes were observed in Dsg3, E-cadherin, and plakoglobin (Pg, also known as γ-catenin), implying a potentially large intracellular store of redundant proteins whose presence masks the detection of protein turnover in this experiment. Collectively, these results suggest that mechanical strain does not necessarily enhance the formation of calcium-independent DSMs but rather causes alterations in protein stability, e.g., Dp, and junction remodeling.

### 2.3. Dsg3 Regulates the Expression and Localization of YAP in Response to Mechanical Loading

YAP has been identified as a mechanosensor [2] with nuclear relocation of YAP being regulated by mechanical strain [36] and substrate stiffness [37]. To elucidate the underlying mechanism of how Dsg3 regulates junction formation induced by mechanical loading, we investigated YAP and phospho-YAP-S127 (pYAP) expression in a time-course experiment in HaCaTs with exposure to the strain for 6 h. In this case, cells were transferred to the stationary state after straining (relaxation) in an incubator, and total lysates were then extracted at various time points at 0, 2, 6, 24, 48, and 72 h, alongside with static control cells (Figure 3A). Consistently, no evident increase was observed in Dsg3 in samples harvested immediately after strain compared to a stationary sample (0 h, Figure 3B). However, a gradual increase in Dsg3 was detected later in post-strained cells, compared to non-strained, from 6 h with a peak at 48 h. Interestingly, in response to cyclic strain, pYAP showed an initial reduction compared to static cells (0 h) and this was followed by a swift recovery within 2 h in post-strained cells, with climbing up to the baseline of static control cells and maintained at this level for up to 72 h of the experiment (Figure 3B). YAP also showed a trend with an increase in post-strained cells. In contrast, in static populations, both pYAP/YAP showed a decline after 24 h and remained at this low level until 24/48 h, indicating their degradation in stationary confluent cultures. Generally, it was observed consistently that these three proteins exhibited a similar shift in their response to mechanical loading, with a gradual increase and stabilization at the baseline compared to static cells in which the activity of pYAP (indicative Hippo) appeared to be transient and was gradually switching off in mature and well-established confluent cultures. The increased levels of pYAP/YAP along with Dsg3 with retardation in post-strained cells, relative to static controls, were confirmed by both the immunostaining and Western blotting analyses.

To examine the connection between Dsg3 and YAP, next, we analyzed protein expression by Western blotting in siRNA pre-treated HaCaTs with/without the cyclic strain that showed marked Dsg3 depletion and concomitant YAP and pYAP (Figure 3C). The nuclear and cytoplasmic fractionation analysis further demonstrated a reduction of both proteins in both subcellular compartments. Furthermore, to validate this finding, a Dsg3 knockdown was performed with additional different siRNAs alongside a different scrambled siRNA control, and again, a similar finding with a reduction of YAP/pYAP was observed in all knockdown cells compared to controls (Figure 3D). Moreover, we subjected the siRNA pre-treated cells to cyclic strain for 6 h before transferring them to a stationary state, followed by extracting lysates at various time points. The resulting Western blots indicated a marked reduction of both YAP and pYAP in cells with Dsg3 depletion, with subtle changes in E-cadherin and steadily rising in Dsg3 in both control and knockdown cells (Figure 3E).

Since both YAP and pYAP were highly soluble to non-ionic detergents, such as Triton, we analyzed their expression in formaldehyde-fixed cells (-Triton) with/without the cyclic strain. YAP staining in control cells displayed predominant cytoplasmic distribution, and the mechanical strain resulted in its elevation (~2-fold, Figure 4A). In contrast, Dsg3 depletion resulted in a marked decrease of YAP with concomitantly enhanced nuclear translocation detectable by this protocol, in response to strain (Figure 4C). Intriguingly, the pYAP staining revealed a sub-pool of proteins located at the cell borders, with an increasing trend in cells exposed to mechanical loading (control cells, Figure 5A). In contrast, the cyclic strain of Dsg3 depleted cells evoked a remarkable reduction of pYAP, with concomitant nuclear translocation, compared to static counterparts (Figure 5B,C). In addition, the biotinylated assay for the surface protein analysis in strained and non-strained cells also indicated enhanced expression for both YAP/pYAP in strained versus non-strained cells (Figure 5D). To verify that the membrane expression is specific in epithelial cells, immunofluorescence for pYAP was performed in various non-epithelial cell types that revealed the membrane pYAP detected exclusively in keratinocytes but not in other cell lines such as oral fibroblasts, 3T3, and Cos-1 cells (Appendix A). Immunostaining in two different siRNA pre-treated cells seeded on coverslips at high and low densities also showed the nuclear retention of pYAP/YAP, with augmented nuclear/cytoplasm ratios, in sparse cultures of Dsg3-depleted cells in contrast to their confluent counterpart or control cells plated at low density (Appendix A). Moreover, a titration of Dsg3-siRNA at a series of concentrations, i.e., 5, 30, and 100 nM, in T8-Vect control and T8-D3 detected a dose-dependent response in pYAP reduction, but with compensation by elevated Dsg3 levels in T8-D3 cells (Figure 4B). In line with enhanced peripheral pYAP in response to strain (Figure 5A,B), increased pYAP was also detected at the cell borders in T8-D3 compared to T8-Vect control, and Dsg3 depletion abolished such pYAP signals at the cell borders with a result of marked nuclear relocation (Appendix A).

Taken together, these results support our hypothesis that Dsg3 regulates YAP/pYAP and is required for their nuclear export and the surface recruitment of pYAP in keratinocytes that might facilitate the AJ junction formation and the onset of CIP.

### 2.4. Dsg3 Colocalizes and Forms a Complex with pYAP, Which Is Sequestered to the Plasma Membrane

Confocal microscopy further demonstrated the colocalization of Dsg3 and pYAP at the plasma membrane in HaCaT, as well as in primary, keratinocytes (Figure 6A). Then, we monitored YAP and pYAP expression profiles alongside Dsg3 in HaCaTs at sub-confluence, newly established-confluence, and over-confluence culture (up to five days) by Western blotting analysis. The expression of pYAP/YAP appeared to be transient and reached a maximum in freshly confluent cultures, similar to Dsg3, before decline when cells reached over-confluence (Figure 6B). E-cadherin, Dp, Plakophilin-1/3 (PKP1/3), and Pg were also analyzed. While E-cadherin and PKP1/3 showed a steady increase, with E-cadherin peaking at the fresh confluence and staying at the same levels for up to five days of the experiment, Dp reached a peak in fresh confluence but then underwent degradation with multiple degraded bands existing in over-confluent culture. Pg, however, showed less change in all aged cultures. Together, these data suggest that the regulation of Dsg3 and pYAP seems to be associated with one another but is independent of other junctional proteins analyzed, including E-cadherin.

Next, co-immunoprecipitation (co-IP) with a specific antibody to pYAP-S127, as well as YAP, was performed in HaCaT lysates extracted from freshly confluent cultures. It was demonstrated that Dsg3 co-immunoprecipitated with pYAP indicating they form a complex, but yet was barely detectable for YAP (Figure 6C). IP with anti-Dsg3 detected a diffuse band of pYAP in conditions with calcium addition in buffers throughout the IP procedures. To confirm such a complex formation in primary keratinocytes, co-IP was repeated in foreskin cells, and in this case, Dsg3, as well as PKP1/3 and 14-3-3, were detected in the complex purified with anti-pYAP IgG and only plakophilin 1 (PKP1) and 14-3-3 were found in the YAP IP (Figure 6D). These findings demonstrated that Dsg3 forms a complex with pYAP and sequesters it to the cell surface via a protein complex containing PKPs.

Double immunostaining for pYAP/Dsg3 in siRNA transfected cells demonstrated the loss of pYAP at the plasma membrane in Dsg3-depleted cells compared to control (Figure 7A). To determine the specific role of PKP1 in this process, we also performed PKP1 knockdown in HaCaTs that revealed PKP1 silencing caused disruption of Dsg3 at the junctions with concomitant loss of peripheral pYAP, as well as significant reduction indicating a dependence of the Dsg3/pYAP interaction on PKP1 (Figure 7B,C). Diffuse cytoplasmic Dsg3 staining with enhanced cell spreading, resembling the Dsg3-depleted cells [38], was also observed in cells with PKP1 depletion. Taken together, these data suggest that Dsg3 connects to pYAP via PKPs (see the model below).

### 2.5. Elevated Expression of Dsg3 in Response to Substrate Stiffness, Similar to other Force Sensors

In addition to mechanical loading, substrate stiffness also has a substantial impact on a wide range of cell behaviors, such as adhesion, spreading, locomotion, differentiation and cell fate decision [8,39]. E-cadherin serves as a mechanosensor in response to substrate stiffness [10,32,40]. To address whether Dsg3 also responds to substrate stiffness, we examined Dsg3 expression, alongside E-cadherin and other well-known force sensors, such as Myosin IIa and α-catenin, in cells seeded on collagen type I coated polyacrylamide hydrogels of varying stiffness (elastic modulus at approximately 8, 70, 215 kPa) that cover the biophysical stiffness of the skin [41], alongside glass coverslips (65 GPa [39]). The results indicated that their expression correlated with substrate stiffness (Figure 8A, Appendix A). As anticipated, staining for YAP revealed enhanced nuclear localization in Dsg3 knockdown cells in a stiffness-dependent manner (Figure 8B,C).

### 2.6. Dsg3 Knockdown Has an Impact on YAP Target Genes

Transcript expression of several YAP target genes, such as *CYR61, CNGF, FOXM1, CMYC, CCNA2,* and *CENPA* [42,43,44] in HaCaT cells with/without Dsg3 knockdown was analyzed and showed a marked reduction in the majority of these genes, including *YAP1* in Dsg3-depleted cells compared to control (Figure 9A). However, cyclic strain did not cause any evident induction of these genes in our current setting (confluent cells harvested immediately after the strain), by qPCR analysis.

### 2.7. Dsg3 Knockdown Has an Impact on Cell Proliferation

Ki67 staining also was performed and indicated a significant reduction in Dsg3 knockdown cells (Figure 9B), but no significant changes were observed in strained and post-strained cells for up to three days, compared to the respective controls. Nevertheless, Western blotting in lysates of strained and non-strained cells revealed an increase in FOXM1 (1.6-fold) and c-Myc (1.9-fold) in post-strained cells at 24 h as compared to the respective static controls (Figure 9C). Importantly, Dsg3 knockdown attenuated such responses in post-strained cells (two-fold reduction in FOXM1 and Cyclin A at 24 h, Figure 9D).

Collectively, these results suggest that Dsg3 plays a role in regulating *YAP1* and its targeted gene expression and cellular function, although the cyclic strain had little effect on cell proliferation in confluent keratinocyte cultures.

## 3. Discussion

This study shows, for the first time, that Dsg3 cooperates with E-cadherin and other characterized mechanosensory proteins, such as α-catenin and nonmuscle actomyosin, in the keratinocyte response to external mechanical forces. Importantly, our findings have identified that Dsg3 serves as an upstream regulator of YAP/pYAP and forms a complex with and sequesters pYAP to the plasma membrane, a process that involves PKPs, to facilitate AJ assembly (Figure 10). A recent report has identified that the desmosomal plaque proteins PKP1/3 bind to 14-3-3γ/σ isoforms that are required in desmosome adhesion [45]. Another independent study has shown that PKPs interact directly with the cytoplasmic tail of Dsg1-3 [46]. This study adds these recent findings and provides direct evidence that Dsg3 is involved in the complex formation with, and sequestering of, pYAP with the targeting of the complex to the keratinocyte surface. This Dsg3/YAP pathway had a positive influence on the expression of YAP target genes associated with cell proliferation. Furthermore, we provide preliminary evidence that oral keratinocytes are more mechanosensitive than cells derived from the skin since many junctional proteins, including both E-cadherin and desmosomal cadherins, were stimulated by mechanical strain; reactions which showed retardation in skin keratinocytes. In summary, this study uncovers a novel signaling role for Dsg3 as a previously unsuspected key player in keratinocyte mechanosensing and mechanotransduction that potentially involves the E-cadherin/α-catenin signaling axis.

Many studies have indicated that AJs, in association with the cytoskeleton, serve as mechanosensors/mechanotransducers and are adaptable to various tension and tugging forces by modifying the cell adhesion strength, size, and protein concentration as well as causing molecular conformational changes at the junctions [7,10,12,13,34,40,47]. Emerging evidence reveals that the extra-junctional Dsg3 cross-talks with E-cadherin and regulates AJ assembly and adhesion [22,23,24,25,38]. This study demonstrated that the loss of Dsg3 action had a major impact on the junction assembly of E-cadherin and Myosin IIa in response to mechanical loading. However, rather than enhancing the cell-cell adhesion strength, we found (based on our current settings) that cyclic strain provoked junction remodeling (accelerated Dp turnover), which is consistent with previous reports indicating negative feedback in response to large scale mechanical stresses [47,48]. Our results also showed variations in response in keratinocytes derived from different body sites. While oral keratinocytes appeared to be more responsive to mechanical stress, exhibiting an instant increase in the expression of a variety of junctional proteins, skin keratinocytes showed a delayed response during the relaxation time period with Dsg3 peak levels achieved later, i.e., 48 h after mechanical loading. Such a delayed response could well reflect a sustained effect after the abrogation of stretch, or a result of “repairing” the monolayer after cyclic strain. Alternatively, this could be due to the mechanical memory of keratinocytes, as reported recently [49,50]. Distinct behaviors and cellular properties between oral and skin keratinocytes have been appreciated, and these include aspects of wound healing and angiogenesis [51,52,53]. Given the distinct expression patterns in Dsg3 staining between oral and skin tissues, we feel this could, in part, reflect their different mechanical environments.

YAP has been identified as a mechanosensor independent of the Hippo pathway, with its nuclear relocation being regulated by mechanical strain and substrate stiffness involving Rho GTPase activity and tension of the actomyosin cytoskeleton [2,4,6,36,37]. Unexpectedly, and apparently, paradoxically, nuclear retention, or retardation of nuclear export, of YAP/pYAP was detected by immunofluorescence in Dsg3 knockdown cells at low cell densities on coverslips but also in confluent cultures in Flexcell wells subjected to cyclic strain. These data suggest a complex situation in which Dsg3 knockdown correlated with YAP nuclear retention but inhibition of target gene expression. We have not examined putative mechanisms, but such inhibition of YAP might be caused by pronounced attenuation when the expression levels fall below the threshold values [2,4] or when the localization of YAP/TAZ is controlled by additional factors, such as cell density and geometry [2,6]. Our previous studies have shown that Dsg3 silencing results in cell morphological changes with flattening due to defects in F-actin organization and cell polarization [38]. Thus the YAP nuclear retention in Dsg3 knockdown cells could be caused, partly at least, by cell flattening that potentiates the extracellular matrix-nuclear mechanical coupling leading to YAP nuclear location [54]. It remains unclear exactly how Dsg3 regulates YAP expression. This could be due to Dsg3’s ability to activate various signal pathways, including tyrosin protein kinase Src, as well as the Fos-binding protein c-Jun, that regulate the actin and actomyosin cytoskeleton [22,23,24,25,38]. Dsg3 depletion has a negative impact on the formation and tension of F-actin [35,52,53] that may attenuate YAP, as supported by our qPCR data indicating a reduction of *YAP1* target gene expression. In our study, however, cyclic strain did not affect the *YAP1* target gene expression nor cell proliferation for up to 3 days in confluent cultures. Nuclear pYAP-S127 location is regulated via S128 phosphorylation of YAP driven by the activation of Nemo-like kinase (NIK) [43] though precisely how Dsg3 silencing impairs epithelial cell proliferation remains unknown [55]. Probably this is due to the downregulation of YAP but this differs from α-catenin knockdown in mammalian cells which activates YAP/TAZ [3,56].

Although pYAP is regarded as an inactive protein, our study suggests a potential role for pYAP in junction formation in keratinocytes in which the cell-cell junctions are particularly abundant. Various junctional proteins, such as α/β-catenins, Crumbs polarity complex, Angiomotins, and Zonula Occludens proteins, are known to physically interact with YAP/TAZ and regulate its activity [5]. This study provides the first evidence that a member of a transmembrane protein and cadherin superfamily, Dsg3, can physically interact with a protein complex containing pYAP/14-3-3/PKP1/3. As illustrated in Figure 10, the expression of Dsg3 is stimulated by mechanical cues from cell surroundings, and Dsg3 binds the pYAP protein complex in the cytoplasm and sequesters it to the plasma membrane to facilitate the junction formation. Thus, we observed isochronous expression profiles of YAP, pYAP, and Dsg3 during the course of mechanically induced activation of the Hippo-pYAP pathway in the post-strained cells or the transient onset of Hippo-pYAP in cells at steady state, grown from sub-confluence to over confluence. In contrast, other junctional proteins such as E-cadherin, Dp, PKPs, and Pg did not show the same expression pattern. Dsg3 depletion caused a failure in strain-induced pYAP activation or recovery. The Hippo pathway is a prerequisite for contact inhibition of cell proliferation (CIP) that controls tissue homeostasis and organ growth [1,2]. CIP is regarded as a classical paradigm of epithelial biology, and its loss contributes to cancer development and progression. The previous study identifies that α-catenin is a central component in CIP and a key suppressor of YAP [3]. This study provides evidence of a novel pathway of Dsg3/YAP that differs from α-catenin in regulating the Hippo-YAP signaling. Accumulated evidence points to Dsg3 acting as a signaling molecule, and this study connects its signaling role to control keratinocyte proliferation, and therefore organ size, via the mechanism of regulating YAP [2,4].

Notably, two independent reports recently demonstrated that the desmosomal proteins, Dsg2, and Dp function as the biosensor in human cardiomyocytes and canine kidney simple epithelial cells [16,17]. These findings, along with the current study, collectively underscore the importance of desmosomes as mechanosensory and load-bearing structures that coordinate with AJs in control of tissue integrity and homeostasis.

## 4. Materials and Methods

### 4.1. Antibodies

The following mouse and rabbit monoclonal/polyclonal antibodies (Abs) were used: 5H10, mouse Ab against N-terminus of Dsg3 (sc-23912, Santa Cruz, Dallas, TX, USA); H-145, rabbit Ab against C-terminus of Dsg3 (sc-20116, Santa Cruz, Heidelberg, Germany); 33–3D, mouse IgM against Dsg2 (gift from Professor Garrod); rabbit Ab to Dsc2 (610120, Progen, Heidelberg, Germany); Dsc3-U114, mouse Ab to Dsc3 (65193, Progen); PG 5.1, mouse Ab to Plakoglobin (65015, Progen); 115F, mouse Ab to Desmoplakin (gift from Professor Garrod); H-300, rabbit Ab to Desmoplakin (sc-33555, Santa Cruz); 5C2, mouse Ab to plakophilin1 (Progen); PKP3, mouse Ab (ab151401, Abcam, Cambridge, UK), HECD-1, mouse anti-N-terminus of E-cadherin (ab1416, Abcam); 6F9, mouse anti-β-catenin ascites fluid (C7082, Sigma-Aldrich, Dorset, UK); rabbit Ab to α-catenin (ab2981, Abcam); H-90, rabbit Ab to p120 (sc-13957, Santa Cruz); mouse Ab to β-actin (8H10D10, Cell Signaling Technology); mouse Ab to Phospho-Myosin Light Chain 2 (Ser19) (3671S, Cell Signaling Technology, Leiden, Netherlands); mouse Ab to K14 (gift from Professor Leigh); Alexa Fluor 488 conjugated phalloidin for F-actin (A12379, ThermoFisher Scientific); Alexa Flour 488 conjugated rabbit Ab to non-muscle myosin IIa (ab204675, Abcam); D8H1X, rabbit Ab to YAP (D8H1X-XP, Cell Signaling Technology); EP1675Y, rabbit Ab to YAP1 (phosphor S127) (ab76252, Abcam); C-20, rabbit Ab to FOXM1 (sc-502, Santa Cruz); H-432, rabbit Ab to Cyclin A (sc-751, Santa Cruz); PC10, mouse Ab to PCNA (sc-56, Santa Cruz); Y69, rabbit Ab to c-Myc (ab32072, Abcam); 8H10D10, mouse Ab to beta actin (3700S, Cell Signaling Technology); mouse Ab to 14-3-3 gamma (Santa Cruz); normal rabbit IgG (2729S, Cell Signaling Technology); purified mouse IgG1(401401, Biolegend, London, United Kingdom); 14C10, rabbit Ab to Glyceraldehyde-3-phosphate dehydrogenase (GAPDH)-Loading control (14c10, Cell Signaling Technology); B-6, mouse Ab to heat shock cognate 71 kDa protein (Hsc70)-Loading control (sc-7298, Santa Cruz); anti-Lamin A antibody (ab26300, Abcam); Secondary Abs were anti-mouse/rabbit IgG peroxidase antibody produced in goat (A0168/A6667; Sigma-Aldrich); Alexa Fluor 488 goat anti-mouse/rabbit IgG (A11029/A11034; Invitrogen, Dartford, United Kingdom), and Alexa Fluor 568 goat anti-mouse/rabbit IgG (A11031/A11036; ThermoFisher Scientific).

### 4.2. Cell culture and siRNA Transfection

Various epithelial cell types were used in the study; HaCaT immortalized human skin keratinocyte line, SqCC/Y1 human oral buccal squamous cell carcinoma (SCC) line, a cutaneous squamous cell carcinoma T8 cell line (gift from Professor Harwood) with the transduction of empty vector control (Vect Ct) and full length Dsg3 (FL) and primary keratinocytes derived from foreskin and breast skin and oral mucous tissues. HaCaT cell line was cultured in Dulbecco’s Modified Eagle Medium (DMEM; Lonza, Basel, Switzerland) supplemented with 10% fetal calf serum (FCS; 0200850, First Link (UK) Ltd., Wolverhampton, UK) and SqCC/Y1 cells and primary keratinocytes were routinely maintained in EpiLife medium (with 60 mM calcium concentration) (MEPI500CA) supplemented with Human Keratinocyte Growth Supplement (HKGS; ThermoFisher Scientific). For the experiments, the media for SqCC/Y1 and primary cultures were replaced with modified Keratinocyte Growth Medium (KGM) (DMEM: Ham’s F12 = 3:1 supplemented by insulin (5 µg/mL), hydrocortisone (0.4 µg/mL), plus 10% FCS). T8-Vect Ct and D3 cell lines were cultured in KGM all the time. A siRNA sequence specific for human Dsg3 mRNA, corresponding to nucleotides 620 to 640 of the respective coding region (Accession: NM_001944.1) (AAATGCCACAGATGCAGATGA) was designed, and this sequence was subjected to a BLAST database search prior to being synthesized by Dharmacon Research (Colorado, USA) [24,55]. All the other siRNA sequences were purchased from Dharmacon(ON-TARGETplus siRNA-2 J-011646-05, siRNA-3 J-011646-06, and siRNA-4 J-011646-08, Human DSG3, NM-001944), plus two scrambled controls, one was a randomized version of the Dsg3 siRNA sequence (AACGATGATACATGACACGAG), and another, a randomized Dp siRNA (Scram-2: AACAGCGACTACACCAATAGA), all were synthesized and provided by the same company [24,55,57]. PKP1 siRNA (ON-TARGETplus PKP1 siRNA J-012545-05-0002 was purchased from Dharmacon. Transient transfection with scrambled and Dsg3 specific siRNAs at the 100 nM concentration was conducted using the protocol as previously described [24,26,55]. For the generation of stable T8 Vect Ct and FL lines, the routine procedures were used as described previously [23].

### 4.3. Application of Cyclic Strain

The regimen for the cyclic strain was adapted from a previous publication [58]. Briefly, HaCaT and SqCC/Y1 cells, with/without Dsg3 knockdown, were plated and grown to confluence for 1~2 days on collagen-coated BioFlex 6-well culture plates with flexible silicone elastomer bottoms (BF-3001C, Flexcell^®^ International Corporation, Burlington, NC, USA). Each plate was placed over the loading station containing six planar faced posts. Cell monolayers were subjected to equiaxial cyclic mechanical stretching with cyclic strain range of amplitude in 10–15% and a frequency of 5 Hz, in a Flexcell FX-5000 Tension System (Flexcell International for varying durations, i.e., 4–6 h or 24 h. Control cells were seeded in the same BioFlex plates along with the strained cells but maintained at a static state (stationary) without any exposure to mechanical stretch. Cells in the plates after strain were fixed with 3.6% formaldehyde for 10 min and permeabilized in Triton X-100 (0.1% in PBS), or without Triton permeabilization, which preferentially detects peripheral proteins, including those at the junctions [23], before immunofluorescence and fluorescent microscopy. Alternatively, lysates were extracted either immediately after a strain or transferred to static state and harvested later at different time points, for mRNA analysis by RT-qPCR or protein analyses by Western blotting.

### 4.4. Calcium-Independent Desmosome Analysis

Cells seeded in BioFlex plates/Collagen I and subjected to cyclic strain or no strain. After that, cells were washed in calcium and magnesium-free Hanks’ balanced salt solution (14175053, HBSS, Gibco, Dartford, United Kingdom) briefly before being incubated in calcium-free medium (21068-028, calcium-free DMEM (Gibco) supplemented with 10% decalcified FCS plus 3 mM EGTA for 90 min, following the established protocol [35]. Control samples were cells incubated in growth medium with normal calcium concentration. Finally, all samples were fixed with 3.6% formaldehyde for 10 min only before processing for Dp immunofluorescence. The fluorescent staining of peripheral Dp was quantitated in ImageJ (NIH, Maryland, Washington D.C., USA).

### 4.5. Immunofluorescence, Microscopy and Image Analysis

Cells grown either in BioFlex plates or on coverslips were washed briefly with PBS before fixation with 3.6% formaldehyde for 10 min at room temperature. Then samples were either subjected to permeabilization with 0.1% Triton for 5 min before antibody staining or proceeded straightaway for immunostaining without treatment with Triton, which preferentially detects peripheral protein, including junctional. The nonspecific binding sites in samples were blocked for 15–30 min with 10% goat serum in washing buffer before the primary and secondary antibody incubations, each for 1 h at room temperature, respectively. For F-actin and Myosin IIa staining, Alexa Fluor 488 conjugated phalloidin or anti-Myosin IIa was incubated together with the secondary antibodies. All antibodies were diluted in 10% goat serum (G9023, Sigma-Aldrich). Coverslips were washed three times with washing buffer (PBS containing 0.2% Tween 20) after each antibody incubation. Finally, coverslips were counterstained with 4′,6-diamidino-2-phenylindole (DAPI) for 8–10 min before being mounted on microscope slides. Images of fluorescent staining were acquired with a Leica DM4000 epi-fluorescence microscope or confocal Zeiss710/ 880. All images were analyzed with the ImageJ software using the default threshold algorithm for each channel across different samples/condition. Finally, total immunofluorescence intensity (IMF) per cell was calculated as the integrated density per channel divided by the total cell number (measured by the number of Nuclei in the DAPI channel). The data were normalized against the non-strained control in each experiment. For analysis of cytoplasmic and nuclear staining, the IMF intensity in each compartment was measured separately. This was achieved by subtracting the binary image of the nuclei from the channel of interest. The IMF was then measured on these new images with cytoplasmic signals only. The nuclear signal was calculated by subtracting the total IMF by the cytoplasmic signal from each channel. The IMF intensity per cell for each compartment was calculated as described above. For the peripheral protein staining, the nuclear mask was expanded using the ‘dilate’ tool. The remaining procedures were the same as described above.

### 4.6. Preparation and Functionalization of Polyacrylamide (PA) Gels

PA gels were prepared and modified from the previous work [59,60]. Briefly, coverslips were cleaned by sonication in ethanol and air-dried overnight. On the following day, all coverslips were incubated with a 5% 3-aminopropyltrimethoxysilane solution (281778, Sigma-Aldrich) in water for 30 min, followed by another 15 min incubation on a 0.5% Glutaraldehyde solution (A17876, Alfa Aesar, Lancashire, United Kingdom) in PBS, both under agitation. Meanwhile, PA solutions were prepared with a ratio variation between 40% polyacrylamide (AAm) (1610140, Biorad, Hertfordshire, United Kingdom) and 2% bis-acrylamide (Bis) (1610142, Biorad) which resulted in different modulus of the PA gels. In this study, we used three gel formulae; AAm 7.5%, 12%, 12%, and Bis 0.05%, 0.145%, 0.45%, respectively. Therefore, through micro-indentation tests, the results for the modulus of PA gels were approximately 8, 70, and 215 kPa respectively. Once prepared, the gel precursor solution was degassed for 5 min before gelation. To initiate gelation, 6 µl of 10% ammonium persulfate (APS; A3678, Sigma-Aldrich) and 4 µl of fN, N, N′, N′-Tetramethylethylenediamine accelerator (TEMED; T9281, Sigma-Aldrich) were added to 1 mL of the gel precursor solution. Six 10 µl drops of the final mixture were placed onto a Sigmacote^®^ microscope slide. The activated coverslips were quickly placed on top of each drop, allowing them to flatten while adhering to the glass. Gels were left to polymerize at room temperature for 5 min. Afterward, the gels and coverslips were flooded with a 50 mM HEPES solution (H3375, Sigma-Aldrich), gently pushed aside to the edge of the microscope slide and lifted from the surface. The coverslips were then stored in the same 50 mM HEPES solution until use. PA gels were functionalized through incubation with 0.2 mg/mL solution in water of the cross-linker Sulfo-SANPAH (2324-50, Biovision, CA, USA) and double 5 min irradiation by 365 nm light. This allowed the photoactivation of the cross-linker. Gels were washed with both deionized water and PBS. Lastly, gels were incubated with a 50 µg/mL collagen type I solution in PBS (354236, Corning, Wiesbaden, Germany) at room temperature for 3 h. Prior to cell seeding, gels were sterilized with UVB irradiation and a 70% ethanol solution followed by a 30 min media incubation.

### 4.7. Western Blotting, Co-immunoprecipitation, and Biotinylation Assay

Total cell extraction, Triton-soluble, and -insoluble fractionations, Western blotting, and co-immunoprecipitation were performed following the protocols as described previously [23,24]. Protein concentrations in all samples were determined by Bio-Rad *DC* protein assay as a routine. For Western blotting, 5–10 µg of total proteins was resolved by SDS–PAGE and transferred to nitrocellulose membrane before antibody incubations. Equal loading was confirmed before probing for the target proteins in each set of samples. For co-immunoprecipitation (co-IP), ~1 mg protein lysate per sample was used to immunoprecipitate YAP, pYAP and Dsg3 associated protein complexes, respectively, using 3–5 µg of primary antibody against each protein and protein-G magnetic Dynabeads (10003D, Invitrogen), and incubated overnight at 4 °C on rotation. Finally, after washing thoroughly (4× in RIPA buffer and 1× in TTBS), the precipitate was re-suspended in 2x Laemmli sample buffer and heated for 3 min at 95 °C before being resolved by SDS-PAGE followed by Western blotting procedures. For Dsg3 IP, calcium ion at the final concentration of 1.8 mM was added into the RIPA and washing buffers throughout the co-IP procedures. The extraction of surface proteins in strained and non-strained cells was performed by following the instructions provided by the Pierce Cell Surface Protein Isolation Kit (ThermoFisher Scientific).

### 4.8. Reverse Transcription Absolute Quantitative RT-PCR (qPCR)

mRNA harvested using Dynabeads mRNA Direct kit (Invitrogen) was converted to cDNA using the qPCRBIO cDNA Synthesis kit (#PB30.11-10, PCRBIO Systems, London, UK), and the cDNA was diluted 1:4 with RNase/DNase free water and stored at -20 °C until used for qPCR. Relative gene expression qPCR was performed using qPCRBIO SyGreen Blue Mix Lo Rox (#PB20.11-50, PCRBIO Systems, UK) in the 384-well LightCycler 480 qPCR system (Roche, Basel, Switzerland) according to our well-established protocols [61] which are Minimum Information for Publication of Quantitative Real-Time PCR Experiments (MIQE) compliant [62]. Briefly, thermocycling begins with 95 °C for the 30 s prior to 45 cycles of amplification at 95 °C for 1 s, 60 °C for 1 s, 72 °C for 6 s, and 76 °C for 1 s (data acquisition). A ‘touch-down’ annealing temperature intervention (66 °C starting temperature with a step-wise reduction of 0.6 °C/cycle; eight cycles) was introduced prior to the amplification step to maximize primer specificity. Melting analysis (95 °C for the 30 s, 65 °C for 30 s, 65–99 °C at a ramp rate of 0.11 °C/s) was performed at the end of qPCR amplification to validate single product amplification in each well. The relative quantification of mRNA transcripts was calculated based on an objective method using the second derivative maximum algorithm (Roche). All target genes were normalized using a stable reference gene (POLR2A). The primer sequences for all genes analyzed in the study are listed in Appendix A.

### 4.9. Ki67 Staining

The cells were fixed with 2% paraformaldehyde for 30 min at room temperature followed by permeabilization with 0.1% Triton for 5 min. Cells were stained with the anti-Ki-67 (M7240, Dako, Stockport, United Kingdom) and counterstained with DAPI. The proliferation rate was assessed by determining the Ki-67-expressing nuclei in relation to the total number of cells defined by DAPI staining [36].

### 4.10. Statistical Analysis

Statistical differences between controls and test groups were analyzed using unpaired, two-tailed Student’s t-test in all cases. Data are presented as mean ± S.D. unless otherwise indicated. *p* values of less than 0.05 were considered statistically significant. Experiments were usually repeated three times. For image quantitation, all images were routinely acquired in 4–6 arbitrary fields per sample with a Leica DM4000 Epi-Fluorescence Microscope (upright) and analyzed with ImageJ software (NIH). For Western blots, the band densitometry of each blot was normalized against the loading control in the same lane, and the comparison between control and test group was normalized against the control (set as 1) and expressed as a fold change.

## Figures and Tables

**Figure 1 ijms-20-06221-f001:**
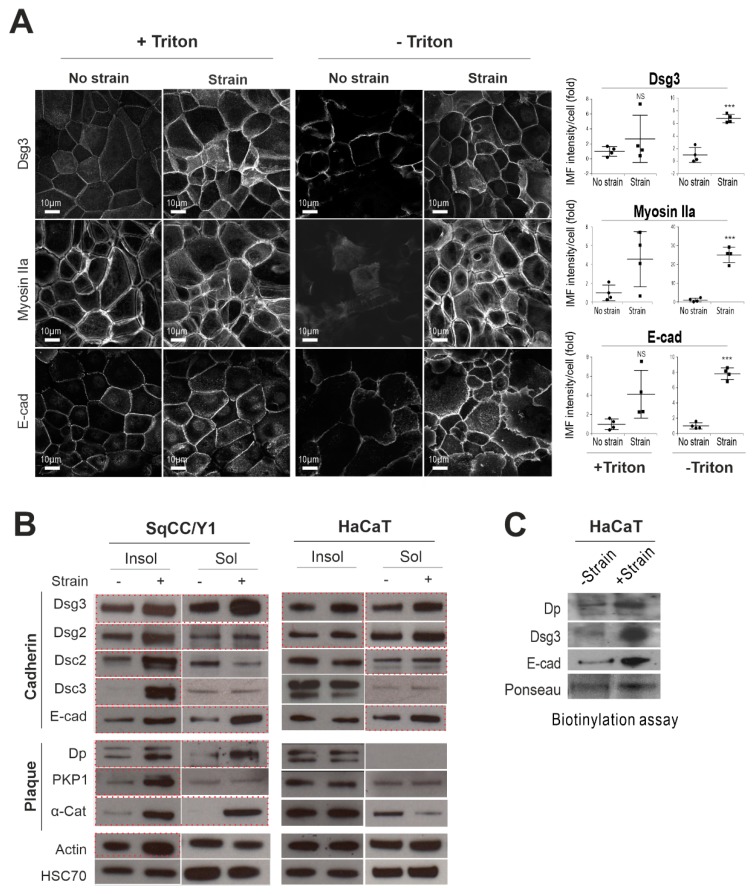
Cyclic strain causes enhanced Dsg3 and E-cadherin expression and their surface assembly in keratinocytes. (**A**) Confocal microscopy of SqCC/Y1 cell line plated in Flexcell wells at confluent density for one day before being subjected to cyclic strain or no strain, for 6 h. Cells were fixed with formaldehyde and then permeabilized with Triton X-100 (+Triton) or no permeabilization (-Triton) before immunostaining for the indicated proteins. Quantitation for each protein is shown on the right (*n* > 4, mean ± S.D., NS: no significance, * *p* < 0.05, *** *p* < 0.001). Scale bar is 10 µm. (**B**) Western blotting analysis for junctional proteins in subcellular fractions, i.e., Triton-soluble (Sol) and insoluble pools (Insol), in oral SqCC/Y1 and skin HaCaT cell lines subjected to no strain (−) and strain (+) for 6 h. HSC70 was used as a loading control. Those marked by the red dotted boxes indicate an evident increase in the expression following strain relative to static cells in either fraction, respectively. (**C**) Biotinylated assay for surface protein expression in non-strained and strained HaCaTs. Cells were seeded in Flexcell wells for one day before being subjected to strain or no strain (5 h). Ponceau staining was used as a loading control here.

**Figure 2 ijms-20-06221-f002:**
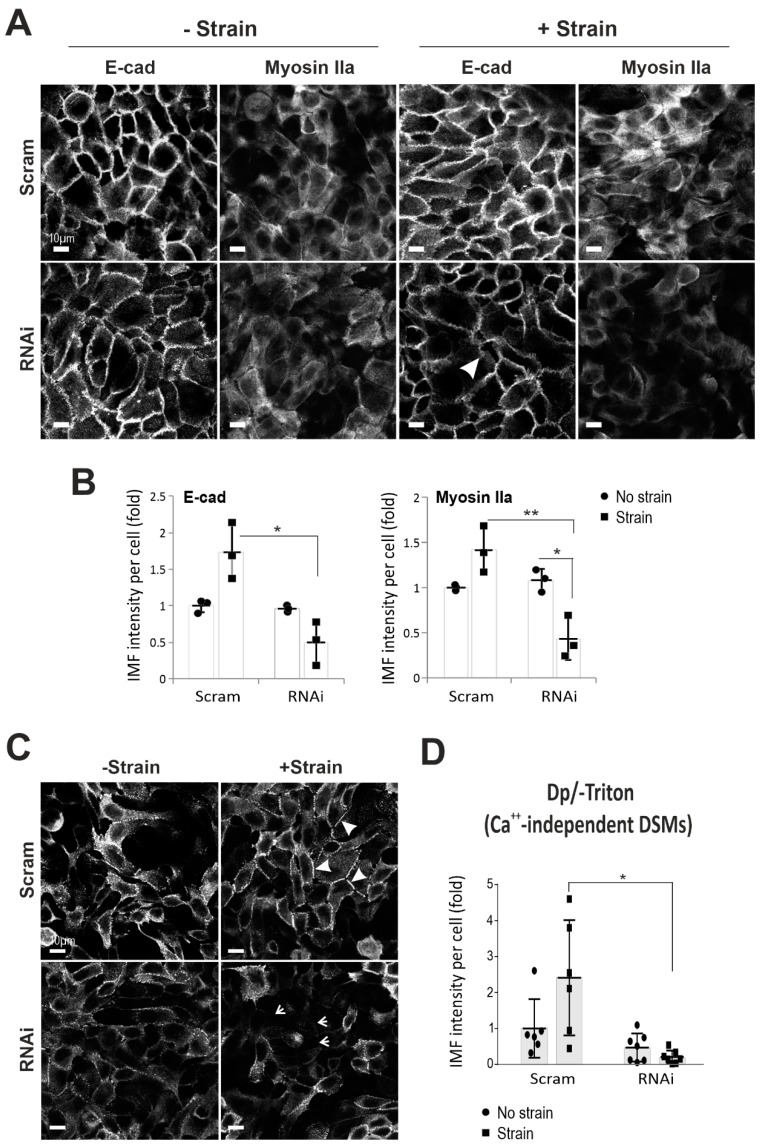
Dsg3 knockdown causes a defect of junction assembly in E-cadherin and Myosin IIa, as well as the calcium-independent desmosomes. (**A**) Confocal images of E-cadherin (E-cad) and Myosin IIa staining in spontaneously immortal HaCaT keratinocyte cells subjected to strain (+Strain) and no strain (-Strain), without (Scram) and with Dsg3 knockdown (RNA interference: RNAi), respectively. Cells were strained for 4 h before being transferred to the stationary state, without changing the medium, for 90 min prior to fixation and immunostaining. (**B**) Image quantitation for E-cadherin and Myosin IIa staining acquired with an Epi-fluorescent microscope (*n* = 3, mean ± S.D., * *p* < 0.05, ** *p* < 0.01). (**C**) Confocal images of HaCaTs subjected to strain or no strain for 4 h. Cells were transferred to the stationary state after the strain and incubated with a calcium-free medium for 90 min in an incubator prior to fixation (formaldehyde) and immunostaining for Dp, an established protocol for the analysis of calcium-independent DSMs [28]. Arrowheads indicate the linear Dp staining at the junction, and arrows show cells missing the calcium-independent DSMs. (**D**) Image quantitation of the Dp staining (*n* = 5–7, mean ± S.D., pooled from two experiments, * *p* < 0.05). Scale bar is 10 µm.

**Figure 3 ijms-20-06221-f003:**
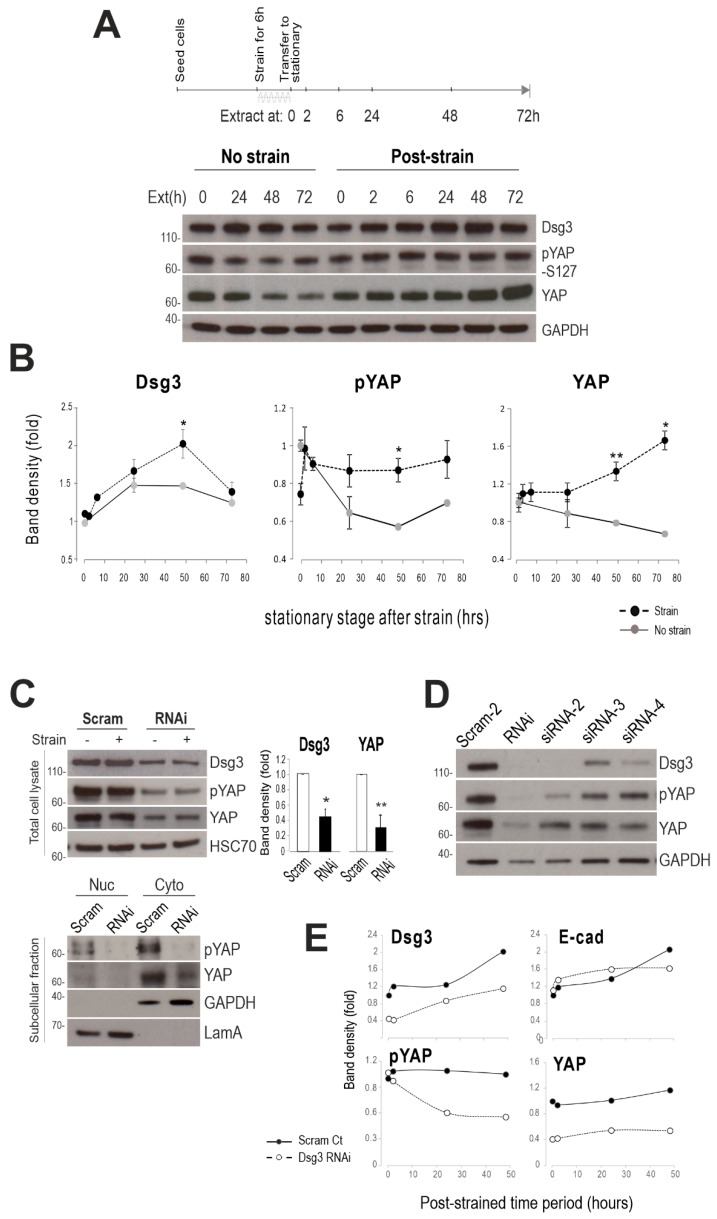
Expression of yes-associated protein (YAP) and phospho-YAP (pYAP) appeared to coincide with Dsg3 in response to cyclic strain. (**A**) Western blotting for HaCaT cells with strained, or non-strained, for 6 h. Lysates were extracted either immediately (0 h) after the strain or later, after being transferred to a stationary state, at various time points as indicated. The timeline for this experiment is shown above the blots. Glyceraldehyde 3-phosphate dehydrogenase (GAPDH) was used as a loading control. (**B**) Band densitometry analysis for Dsg3, pYAP, and YAP expression. Note that all three proteins showed increased expression in response to cyclic strain compared to non-strained samples. Quantitation for each protein was normalized against the loading in each lane and presented as a fold change of the non-strained sample at the 0 h time point (*n* = 3, mean ± S.E.M, **p* < 0.05, ***p* < 0.01). (**C**) Western blots of the indicated proteins in HaCaTs transfected with scrambled (Scram) or Dsg3 specific siRNA (RNAi) for two days. Below were the blots of pYAP and YAP in the nuclear versus cytoplasmic fractionations. GAPDH and LamA were used as loading controls. The bar charts on the right show the band densitometry for Dsg3 and YAP blots (*n* = 3, mean ± S.D.). (**D**) Western blots of HaCaTs transfected with various Dsg3 siRNA sequences (siRNA-2–4: used for the majority of the experiments) alongside a second scrambled control siRNA (Scram-2). GAPDH was as a loading control. (**E**) Band densitometry analysis for the indicated protein expression in Scram and RNAi cells analyzed by Western blotting, and in this case, all siRNA pre-treated cells were subjected to cyclic strain.

**Figure 4 ijms-20-06221-f004:**
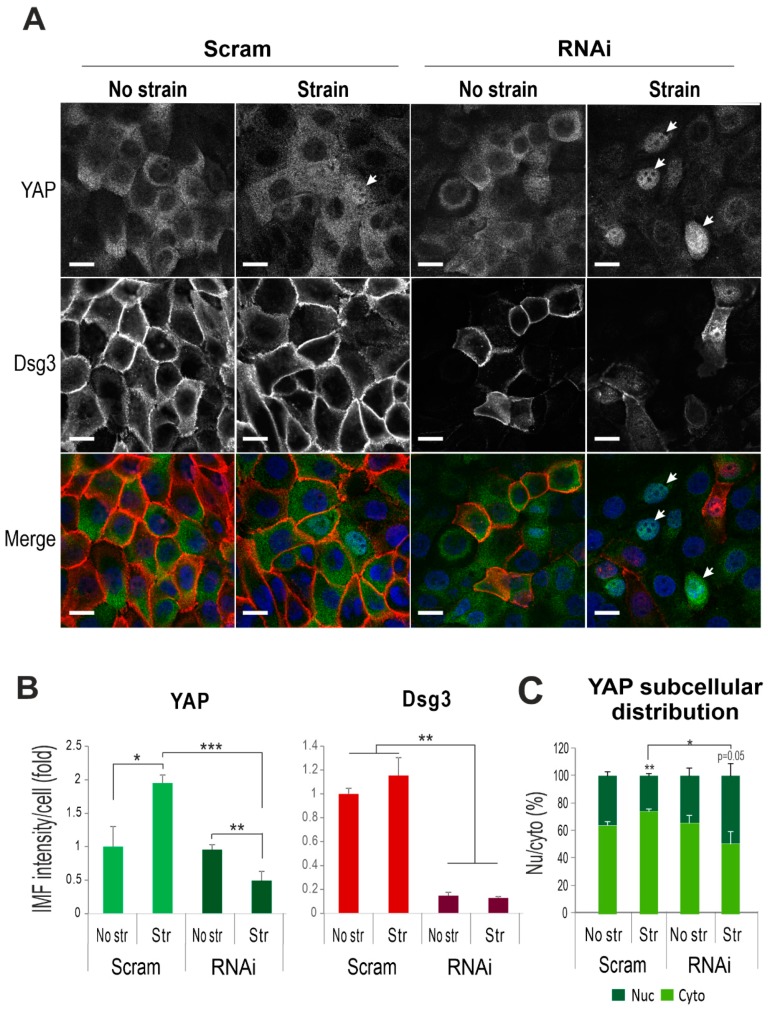
Dsg3 regulates YAP expression and subcellular distribution in response to cyclic strain. (**A**) Confocal images of HaCaT cells, without (Scram)/with Dsg3 knockdown (RNAi), double-labeled for Dsg3 (red) and YAP (green). Cells were seeded at confluent density in BioFlex wells for one day before being subjected to cyclic strain for 6 h or maintained in the stationary state prior to fixation with formaldehyde. Arrows indicate YAP nuclear staining. Enhanced YAP nuclear signals were shown in RNAi treated cells with strain in contrast to other samples in which YAP had predominant cytoplasmic staining. Scale bar, 10 µm. (**B**) Image quantitation. (**C**) Subcellular distribution analysis for YAP. (*n* = 4, mean ± S.D., * *p* < 0.05, ** *p* < 0.01, *** *p* < 0.001).

**Figure 5 ijms-20-06221-f005:**
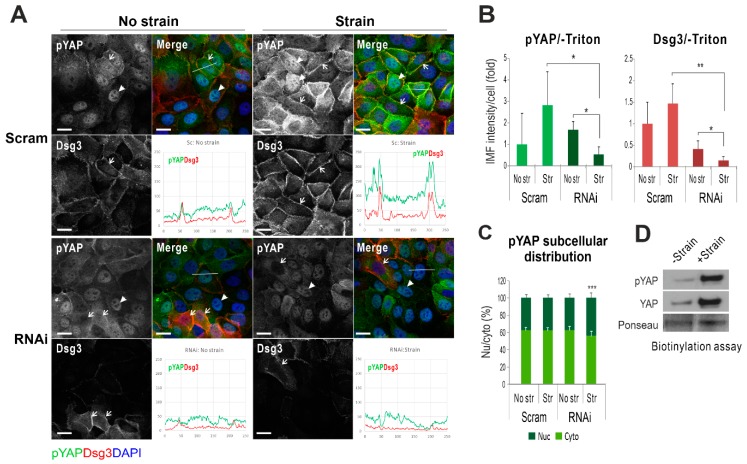
Dsg3 regulates peripheral pYAP expression and localization in its response to cyclic strain. (**A**) Confocal images of HaCaT cells doubled labeled for Dsg3 (red) and pYAP (green). The siRNA pre-treated cells (Scram: control siRNA; RNAi: Dsg3 specific siRNA) were seeded at confluent density in BioFlex wells for one day before subjected to cyclic strain for 6 h or maintained in a stationary condition prior to fixation with formaldehyde only and then immunostained for the indicated proteins. Arrowheads indicate pYAP nuclear staining, and arrows indicate the cells which had Dsg3 expression with concomitantly less nuclear pYAP signals. Note the enhanced colocalization of pYAP and Dsg3 was shown in control cells subjected to the strain, and this was largely abrogated in cells with Dsg3 knockdown, in particular with strain (see the line profile in each condition). (**B**) Image quantification for pYAP and Dsg3 expression, and both channels were subject to nuclear extraction before measurement with a high threshold that detected mainly the peripheral signals. (**C**) Subcellular distribution analysis for pYAP. (*n* = 4, mean ± S.D., * *p* < 0.05, ** *p* < 0.01, *** *p* < 0.001). (**D**) Biotinylated assay for surface pYAP/YAP expression in HaCaTs with non-strain or strain for 5 h. Scale bar, 10 µm.

**Figure 6 ijms-20-06221-f006:**
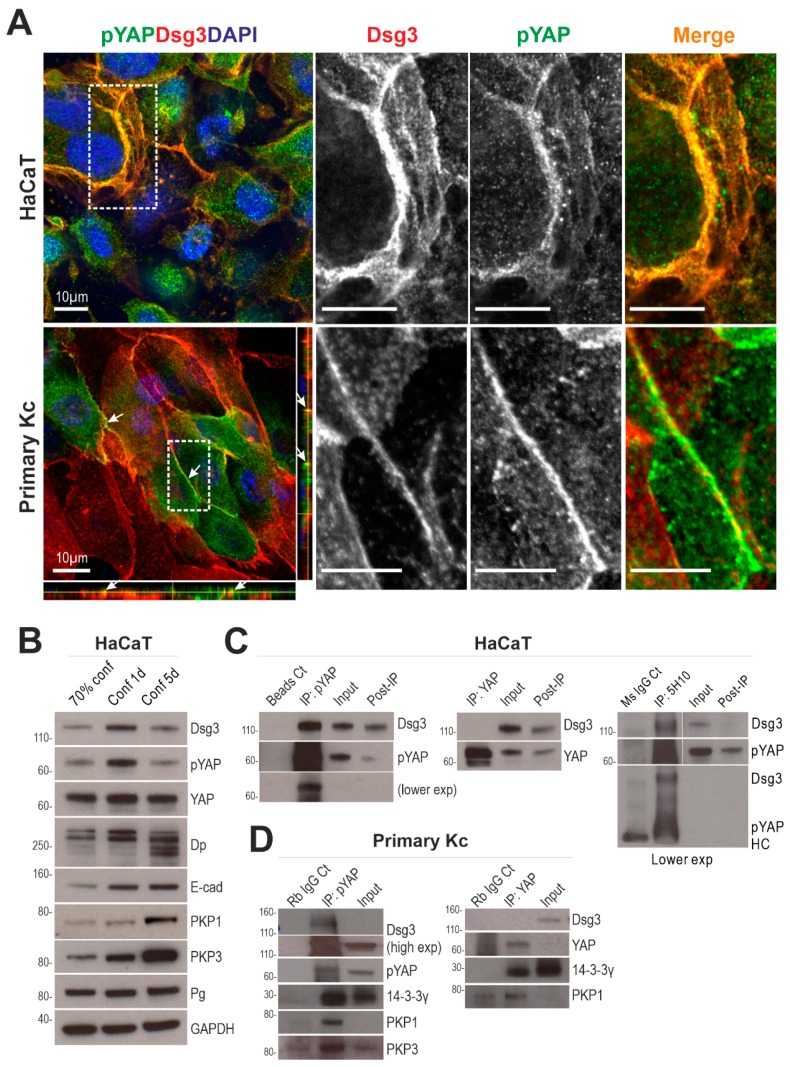
Dsg3 colocalizes and forms a complex with pYAP. (**A**) Confocal super-resolution images of HaCaT and primary keratinocytes double-labeled for Dsg3 and pYAP. The enlarged images of the white dotted boxes with merged and each channel are displayed on the right for each cell line, respectively. Arrows indicate the colocalization of two proteins. Scale bar is 10 µm. (**B**) Western blots of the indicated proteins in sub-confluent (70%), freshly-confluent (100% for 1 d), and over-confluent cultures (5 d) of HaCaT cells. GAPDH was used as a loading control. (**C**) Co-IP analysis of immunoprecipitates in freshly confluent HaCaT cells, purified with anti-pYAP, anti-YAP, and anti-Dsg3 antibodies, respectively. The control lanes included Bead only, mouse IgG alone, input, and post-IP samples. Lower exp: lower exposure. (**D**) Co-IP of immunoprecipitates of primary foreskin keratinocytes, purified with anti-pYAP and anti-YAP antibodies, respectively. The control lanes were rabbit IgG alone and input before IP.

**Figure 7 ijms-20-06221-f007:**
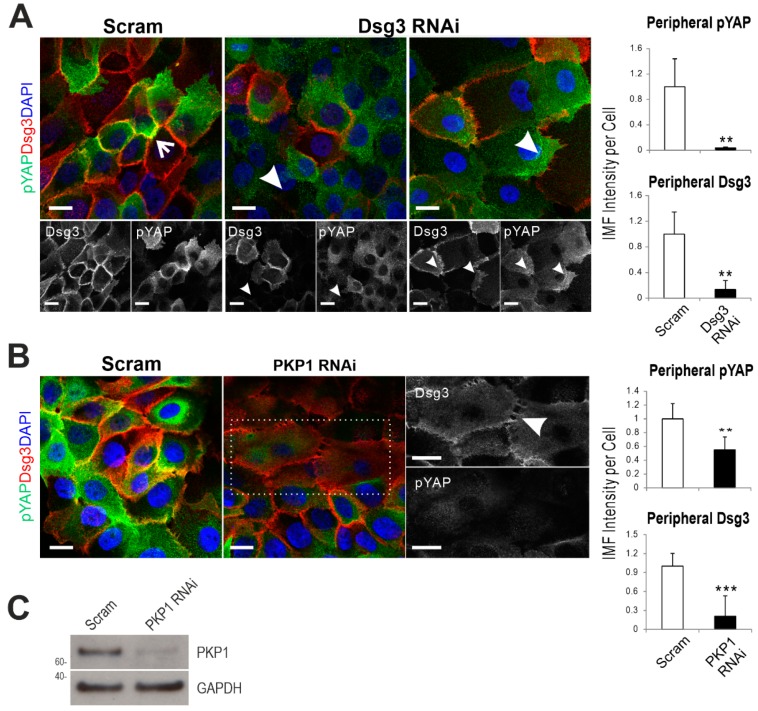
Either Dsg3 or plakophilin 1 (PKP1) knockdown results in a reduction of pYAP peripheral expression. Confocal images of pYAP (green) and Dsg3 (red) staining in HaCaT cells with Dsg3 knockdown (**A**) or PKP1 knockdown (**B**) for two days, respectively. Arrow indicates the colocalization of both proteins at the plasma membrane. Arrowheads in (**A**) indicate the loss of Dsg3 and the concomitant reduction of pYAP in cells and in (**B**) indicates Dsg3 disruption at the junctions with a concurrent loss of peripheral pYAP. Image quantitation’s are shown in the bar charts on the right. The enlarged images of the white dotted box for each channel in (**B**) are displayed on the right. (*n* = 4–6 fields, mean ± S.D., ** *p* < 0.01, *** *p* < 0.001). Scale bar is 10 µm. (**C**) PKP1 knockdown was verified by Western blotting analysis.

**Figure 8 ijms-20-06221-f008:**
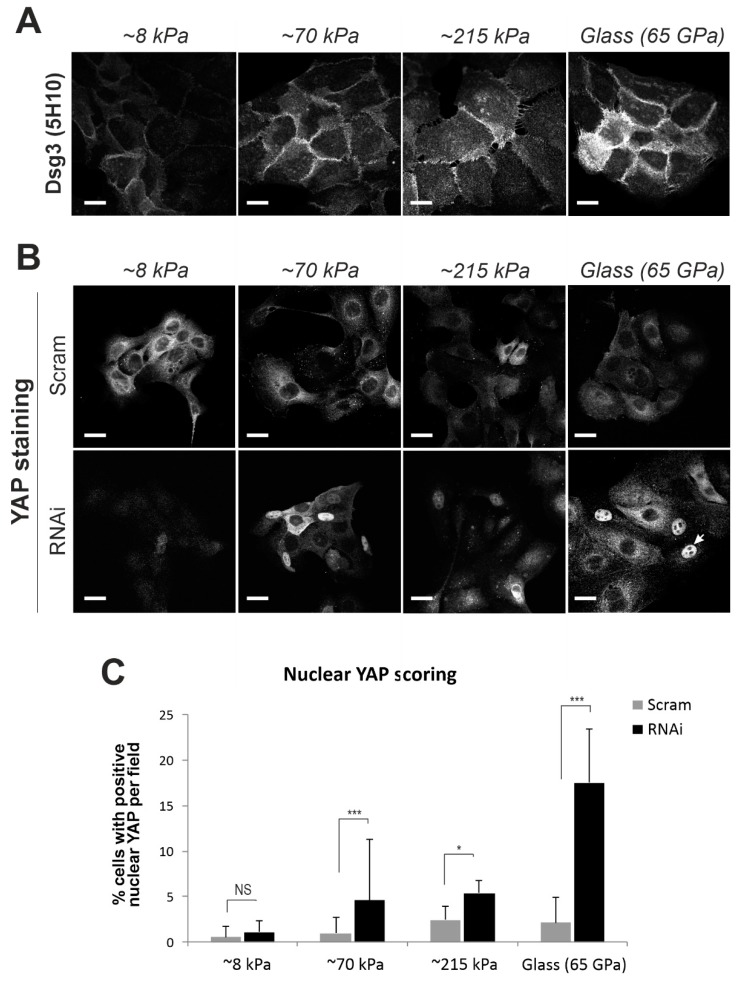
Dsg3 knockdown causes retardation of YAP nuclear exclusion in a substrate stiffness-dependent manner. (**A**) Confocal microscopy of Dsg3 staining in HaCaT cells seeded on collagen type I coated polyacrylamide hydrogels of varying stiffness along with glass coverslip. A correlation between protein expression and substrate stiffness was shown. (**B**) Confocal images of YAP staining in siRNA pre-treated cells seeded on polyacrylamide hydrogels. (**C**) Quantitation for the percentage of cells with positive YAP nuclear staining (*n* > 8, * *p* < 0.05, *** *p* < 0.001, NS: no significance). Scale bar is 10 µm.

**Figure 9 ijms-20-06221-f009:**
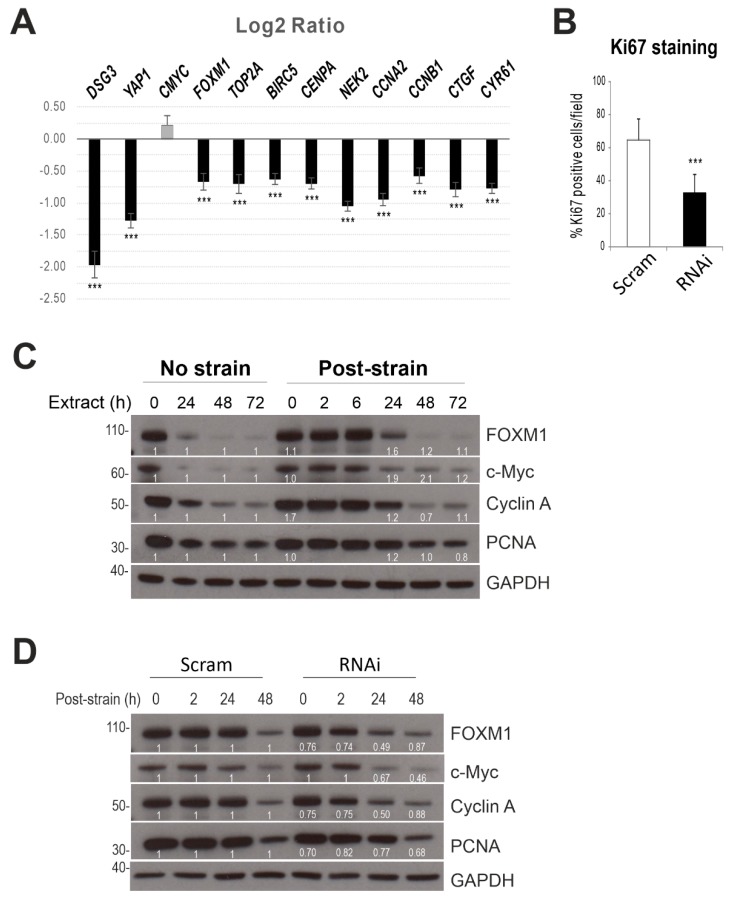
Dsg3 regulates expression of *YAP1* and its target genes as well as other cell proliferation markers in confluent cultures. (**A**) RT-qPCR analysis of cell cycle-associated genes, including YAP targets such as *FOXM1*, *CCNA2*, *CENPA*, *CYR61,* and *CTGF* in HaCaT cells with Dsg3 knockdown. Data are the log2 ratio of RNAi to Scram control (each box was the median of *n* = 6, error bars: S.E.M., *****p* < 0.0001) and are representative of two independent experiments. Except for *CMYC*, all other genes showed a significant difference compared to controls. Cyclic strains showed no effect on these genes in the current setting, i.e., confluent cells subjected to cyclic strains for 6 h before being extracted straightaway for qPCR analysis. (**B**) Ki67 staining in scrambled control siRNA (Scram) and Dsg3 siRNA (RNAi) treated cells (*n* = 6, mean ± S.D., *** *p* < 0.001). (**C**,**D**) Western blotting for the indicated cell proliferation markers in cells subjected to strain/non-strain as well as in siRNA pre-treated cells subjected to cyclic strain for 6 h. GAPDH was used as a loading control. The fold changes in post-strained cells were calculated as opposed to their respective non-strained or Scram control lanes and were displayed underneath each blot.

**Figure 10 ijms-20-06221-f010:**
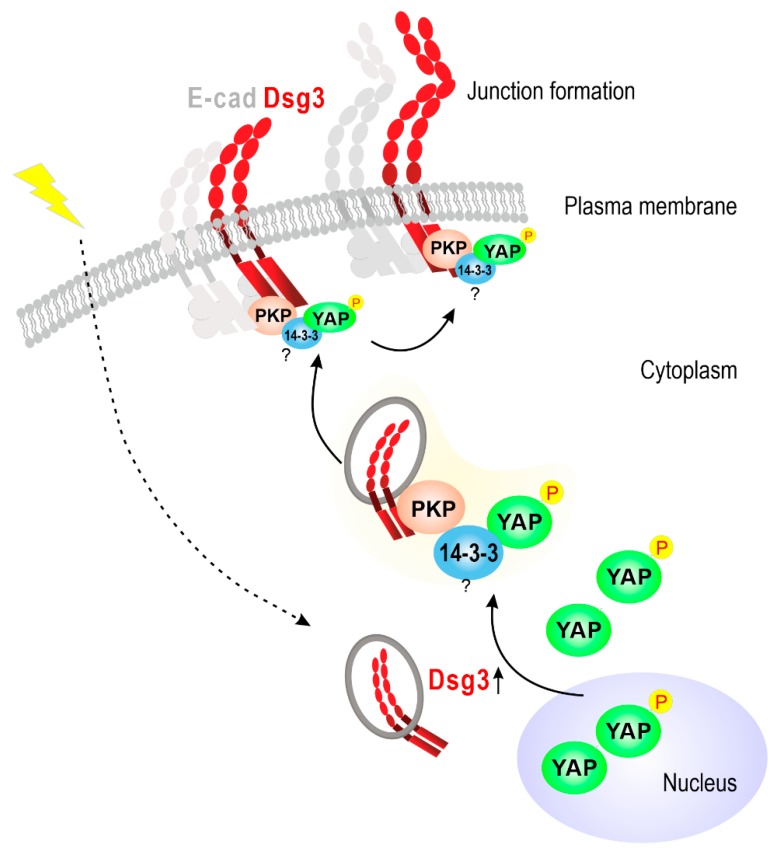
A schematic illustrating that Dsg3 expression is induced by environmental mechanical forces leading to YAP/pYAP nuclear export. Dsg3 preferentially forms a complex with pYAP and sequesters it to the cell surface to facilitate junction formation in keratinocytes. The dotted arrow indicates external mechanical stimulation, the question symbols indicate the lack of direct co-IP data for this protein in this study and arrows indicate positive stimulation of the processes, such as the protein complex formation, surface recruitment and finally, the junction formation.

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
