# Peer review of "Evidence for the Desmosomal Cadherin Desmoglein-3 in Regulating YAP and Phospho-YAP in Keratinocyte Responses to Mechanical Forces"

_ijms, 2019, doi:10.3390/ijms20246221_

Round 1

Reviewer 1 Report

This manuscript is focused on a novel role of Dsg3 as a mechanosensor, which is well written. The results are so interesting but I doubt whether Dsg3 really plays a mechanosensor in skin. The HaCaT cell line is widely used in keratinocyte monolayer culture models. However, the transcriptional expression pattern of cornified envelope-associated proteins is abnormal compared to normal human primary keratinocytes. Furthermore, Dsg3 expression in normal cutaneous keratinocytes is restricted in the basal layer. In discussion, authors mentioned that “Given the distinct expression patterns in Dsg3 staining between oral and skin tissues, we feel this could in part reflect their different mechanical environments”.  I agree their mechanical environments are different but Dsg3 contribution in skin is skeptical.

Queries and recommendation

Abbreviations should be explained at first mention. Please move “Material and Methods” after “Discussion”. Please add city and country names following company name at first appeared. On line 123: Which part of mucosa, keratinized or non-keratinized ones, did you use in this setting? “Ki67 staining and proliferation rate analysis” is suitable for the subtitle. In 2.10, the information of software used for both statics and image analysis should be provided. On line 261-269, 341-345, 412-415, 463-466 and 520-527, authors provided some information or materials and methods at the beginning. Enough experimental conditions should be provided in materials and methods. If necessary, descriptions must be concise. Basically, the above mentioned parts should be deleted. Please check line 197 and 526. Which stiffness of 214 or 215 kPa is correct? References have to be listed according to citation style guide for MDPI journals in order of appearance in the text. Please see the instructions for authors. Why did you compare not HaCat cell line but NTERT cell line with non-epithelial cells in only supplementary Figure S6? Considering the context of your manuscript, HaCaT or SqCC/Y1 cell line looks like to be adequate.

Author Response

This manuscript is focused on a novel role of Dsg3 as a mechanosensor, which is well written. The results are so interesting but I doubt whether Dsg3 really plays a mechanosensor in skin. The HaCaT cell line is widely used in keratinocyte monolayer culture models. However, the transcriptional expression pattern of cornified envelope-associated proteins is abnormal compared to normal human primary keratinocytes. Furthermore, Dsg3 expression in normal cutaneous keratinocytes is restricted in the basal layer. In discussion, authors mentioned that “Given the distinct expression patterns in Dsg3 staining between oral and skin tissues, we feel this could in part reflect their different mechanical environments”.  I agree their mechanical environments are different but Dsg3 contribution in skin is skeptical.

Queries and recommendation

Abbreviations should be explained at first mention.

Amended.

Please move “Material and Methods” after “Discussion”.

Moved.

Please add city and country names following company name at first appeared.

Added.

On line 123: Which part of mucosa, keratinized or non-keratinized ones, did you use in this setting?

SqCC/Y1 cell line is a human buccal SCC line (line 121) and T8 cell line was derived from trunk skin SCC.

 “Ki67 staining and proliferation rate analysis” is suitable for the subtitle.

Corrected (please see line 380).

In 2.10, the information of software used for both statics and image analysis should be provided.

Corrected (line 660).

On line 261-269, 341-345, 412-415, 463-466 and 520-527, authors provided some information or materials and methods at the beginning. Enough experimental conditions should be provided in materials and methods. If necessary, descriptions must be concise. Basically, the above mentioned parts should be deleted.

Please check line 197 and 526. Which stiffness of 214 or 215 kPa is correct?

Corrected (now in line 601).

References have to be listed according to citation style guide for MDPI journals in order of appearance in the text.

Corrected.

Please see the instructions for authors. Why did you compare not HaCat cell line but NTERT cell line with non-epithelial cells in only supplementary Figure S6? Considering the context of your manuscript, HaCaT or SqCC/Y1 cell line looks like to be adequate.

Corrected (please see revised Fig S6 in supplementary).

Reviewer 2 Report

1. Please correct, under section 2.6. Preparation and Functionalization of polyacrylamide (PA) gels, in several positions ul by μl (rows 199, 201)

2. Row 324: please define abbreviation Dsg3 RNAi; it is deductible, but… The same for RNAi used in Figure 2, and some supplemental materials. I guess it means cells with inhibited mRNA for Dsg3.

3. Please explain why, in various experimental contexts, different strain periods of time were used.

4. Figure 10 is a little bit confusing because desmoglein 3 is represented inside the cell as being soluble in cytosol, or in the lumen of intracellular membrane coated structures, but not in the endomembrane of membrane bounded elements, as has to be. I suggest a redesign of this image to be as correct, as suggestive it is.

Author Response

Please correct, under section 2.6. Preparation and Functionalization of polyacrylamide (PA) gels, in several positions ul by μl (rows 199, 201)

Corrected.

Row 324: please define abbreviation Dsg3 RNAi; it is deductible, but… The same for RNAi used in Figure 2, and some supplemental materials. I guess it means cells with inhibited mRNA for Dsg3.

Corrected.

Please explain why, in various experimental contexts, different strain periods of time were used.

We planned the experiments primarily focused on 2 time points, i.e. 6 hours and 24 hours due to limited access to the equipment (9 am ~18 pm) that belongs to The School of Engineering and Materials Science, QMUL in different (Mile End) campus, approximately 30 minutes by bus. Each time, we had to prepare cells before the cyclic strain experiments and harvested samples after that. So the time points of the experiments were slightly varied.     

Figure 10 is a little bit confusing because desmoglein 3 is represented inside the cell as being soluble in cytosol, or in the lumen of intracellular membrane coated structures, but not in the endomembrane of membrane bounded elements, as has to be. I suggest a redesign of this image to be as correct, as suggestive it is.

Corrected.